# Micro-Analytical Study of a Zeolites/Geo-Polymers/Quartz Composite, Dielectric Behaviour and Contribution to Brønsted Sites Affinity

Abdel Boughriet [1,*], Oscar Allahdin [1,2], Nicole Poumaye [1,2], Gregory Tricot [1], Bertrand Revel [3], Ludovic Lesven [1] and Michel Wartel [1]

[1] Laboratoire Avancé de Spectroscopie Pour les Interactions, la Réactivité et l'Environnement, Université de Lille, CNRS, UMR 8516—LASIRE, 59000 Lille, France
[2] Chaire Unesco «Sur la Gestion de l'eau», Laboratoire Hydrosciences Lavoisier, Faculté des Sciences, Université de Bangui, 908 Bangui, Central African Republic
[3] Service RMN, Université de Lille, Bât. C4, 59655 Villeneuve d'Ascq, France
* Correspondence: abdel.boughriet@univ-lille.fr

**Abstract:** The chemical and mineralogical surface properties of a brick-derived composite were examined by using an environmental scanning electron microscopy (ESEM) equipped with an energy dispersive X-ray spectrometer (EDS). Investigations revealed that the material could be assimilated to an adsorptive membrane having zeolites deposited onto quartz matrix. In our calculation, the membrane was considered as a diphase composite and its dielectric constant was evaluated from theoretical models developed in the literature. Electro-kinetic analysis showed that composite surfaces were hydroxylated with the formation of hydroxyl groups which behaved amphoterically. A theory-based approach was used for calculating thermodynamic constants relative to surface-protonation equilibriums. In the H-form of the composite, the occurrence of bridging Si–(OH)–Al sites were evidenced by mathematical calculations utilizing equations in direct relation to mineralogical, crystallographic and dielectric surface characteristics. [1]H MAS NMR spectroscopy confirmed the existence of bridging Brønsted acid sites at acidified composite surfaces interacting with ammonium (as probe ions). Owing to advancements in brick-based composites research, this should lead more to the development of "ceramic" adsorptive membranes with natural clay materials.

**Keywords:** zeolite; composite; hydroxyl groups; protonation equilibrium; dielectric constant; surface characterisation; adsorptive membrane

## 1. Introduction

Zeolites are crystalline alumosilicate microporous materials with high cation exchange capacity. They have frequently been used for removing metal ions and other toxic contaminants from aqueous solutions. However, such inorganic adsorbents are usually fine powders, which are not beneficial for ion diffusion through a fixed-bed column. In contrast, mixing various mineral compounds with variable surface characteristics form composites with heterogeneous behaviour (chemical composition, mineralogy, surface morphology/area/porosity and permeability) that contribute to affect strongly adsorption/filtration processes during the removal of contaminants from aqueous solution [1,2]. Among numerous discussed composites, "zeolites/geopolymers" composites have proved to be promising adsorbents to remove ionic pollutants from wastewater [1,2]. In addition, in these mixed materials geo-polymeric gels act as strong and reliable supports for zeolitic crystals.

Recent studies on modified brick as "exchanger" had received some attention due to its high adsorptive and selective behaviour from very low ionic pollutants concentrations in solution. These works concerned particularly the removal of metallic pollutants from

aqueous solutions like: Hg(II) [3]; Zn(II) [4]; Cr(VI) [5]; both Cr(VI) and Ni(II) [6]; or Ni(II) alone [7]; $Fe^{2+}$, $Cd^{2+}$, $Cu^{2+}$, $Pb^{2+}$, and $Zn^{2+}$ [8–12]. In the lab, we transformed a brick containing metakaolinite (from Central African Republic) into low-silica zeolites by alkaline activation. Synthesized alkali-brick possesses intrinsically a high ability for the adsorption of cationic metals due to its high framework aluminium content and its correspondingly large number of negative framework charges (each $AlO_4$ tetrahedral unit leads to a net negative charge). In addition, the high amount of quartz (60–65 w%) in the synthesized compound contributes as a "matrix" to facilitate the flow of inlet solutions through fixed-bed columns. Such a modified brick can be assimilated to an adsorptive membrane prepared from blended zeolites into quartz matrix [13]. It can also be considered as a surface composite membrane in which the adsorbent (zeolites) locates on the surface and is in direct contact with the solution [14]. The assimilated "brick" membrane then possesses a dual function of adsorption and filtration: with zeolitic aggregates favouring adsorption capacity, while quartz matrix allowing mostly low operating pressure and high permeability flux. In a first objective, we conducted a chemical and mineralogical surface analysis of the modified brick by using an environmental scanning electron microscopy (ESEM) equipped with an energy dispersive X-ray spectrometer (EDS).

In order to avoid too much experimental testing/high cost/long delay, computational simulation with the finite element concept had been employed for exploring/finding better combinations of materials and minimizing failures and negative effects [15–21]. However, before undertaking computational calculations, it is important to know that the selection of zeolitic adsorbent deposited onto quartz matrix depends on their adsorption and desorption performance towards target pollutants in groundwater, which can be examined by batch adsorption tests and fixed-bed column tests for adsorptive-membranes design. Adsorption thermodynamics/kinetics can then be analyzed to study the mass transfer mechanism of pollutants from groundwater to interior membrane and improve the full-scale design of the adsorptive composite membrane. For higher-scale adsorption problems observed in zeolites-based composites such as the effects of minerals type, grain sizes, degree of heterogeneity, surface charges, and electric distribution, the finite element method (FEM) become interesting to be applied to our system (by setting different properties for meshes to simulate related problems) [22–26]. But, in practical applications as the pH of groundwater varies at different contaminated sites, it has an additional effect on column adsorption by affecting chemically both adsorbates (by involving hydrolysis reactions and/or oxides/hydroxides precipitations) and surface adsorbent (by involving hydrogen bonds and hydrolysis complexation). Note that acid effects on NaA zeolite membrane and permeation performance were well revealed by analyzing microstructure evolution with inlet-solution pH using electrochemical impedance spectroscopy [27]. This had led us in the present work to study preliminarily composite acidity (which is dependent upon the number and distribution of Al substitution) and related thermodynamic and structural properties. On the basis of these data and depending upon the intrinsic acid-base characteristics of (chosen) cationic metals in aqueous media, computational (FEM) analyses should be very useful to conceive artificially adaptive zeolites-based composites with appropriate Al/Al+Si) by simulating column experiments and thereby optimize mineralogical membrane composition [22–26]. Indeed, fabrication of such active composite membranes with right Al/(Al+Si) ratios (by performing e.g., dealumination or desilication of zeolite structure [28]) becomes a crucial aspect to make deposited sodic-zeolites have both right adsorptive properties towards specific heavy metals and assure their regeneration by controlling membrane pH and avoiding precipitations (e.g., in the case of $Pb^{2+}$ ions). For instance, adsorptive composite membrane containing zeolitic deposits with maximum Al/(Al+Si) ratio (like that studied here) has been found to be well adapted to the removal and recovering of radioactive elements (like caesium and rubidium) from contaminated waters.

According to models for adsorption of charged species/complexes from aqueous solution onto oxide surfaces [29], total adsorption free energy term accounted for coulombic interaction (usually an attractive term), solvation energy (a repulsive term), and a chemical

free energy term which was employed additionally to compensate the solvation term. On this view, proton solvation occurred electro-statically at the composite-water interface, and corresponding Gibbs free energy of hydroxonium ions at the solid surface depended upon the change in field intensity ($\Delta E$) on moving $H^+/Na^+$ charges from the aqueous solution with a dielectric constant $\varepsilon_{H20}$ towards the solid with a dielectric constant, $\varepsilon_{solid}$. The value of the dielectric constant of the adsorbing solid was therefore extremely important in determining the magnitude of the change in free energy of solvation of the adsorbed ion. Although protons adsorption onto quartz occurred negligibly, the relatively high dielectric contribution of this mineral to solvation free energy term had to be taken into consideration in thermodynamic calculations relative to adsorptive properties of the composite [29]. In a second objective, we then calculated the global dielectric constant of the brick composite from Bruggeman, Lichterecker and Hashin-Shtrikman's equations.

Although the surface characteristics and protons reactions at zeolite/water interface were well examined in the past, until now no intensive study of these properties had been undertaken for a complex natural mineral like a brick which is derived from kaolinite-rich soils (in Bangui) and mainly composed of metakaolinite, quartz, and to a lesser extent, of illite, rutile and iron oxides/hydroxides. To fill this gap, we had initially conducted potentiometric studies in order to investigate the acid–base properties of modified brick/water suspension. Unfortunately, it had been difficult to study electrochemically brick composite for which the interpretation of surface titration results was found to be complex because of lengths of reaction time to achieve steady-state, as already observed for other minerals [30,31]. Observed low-rate phenomena would result from cations sizes and competitive adsorption, implicating water molecules for available surface exchange sites [30,31]. Thereby, low reaction kinetics prevented us to simulate adequately surface protonation processes using parameters derived from experimental data. As an alternative, we instead employed in the present work theoretical methods which made easier prediction of surface protonation constants for complex structures containing a variety of minerals. Also, in order to address the current literature debate about effects of support material on acid strength of an "adsorptive" surface composite membrane, there exists a need to systematically evaluate effective dielectric contribution to the amphoteric behaviour of hydroxylated surface groups. In a third objective, by taking into account the dielectric contribution of quartz matrix we addressed a theory-based approach in the aim to evidence the chemical nature of hydroxyl groups involved in heterogeneous protonation reactions at composite surfaces and to assess corresponding thermodynamic equilibriums. To our knowledge, there are still no so detailed works in the literature about determination of surface-protonation equilibrium constants for "adsorptive composite membranes like that studied here. More generally, such research should be required for adsorptive membranes in the aim to determine effective thermodynamic surface properties of the adsorptive material deposited onto support matrix.

In a fourth objective, we used $^1$H solid-state NMR spectroscopy as a suitable tool for the characterisation/nature of acidic OH groups, and acid strength on partially decationized zeolite frameworks by undertaking investigations upon loading of a probe molecule such as ammonia [32–35].

## 2. Materials and Methods

### 2.1. Brick Characterisation

The raw material used in the experiments was obtained from a brick made locally by craftsmen in Bangui region (Central African Republic).Before use as an adsorbent, several physical/chemical treatments were carried out on the brick. First, it was broken into grains and sieved with sizes ranging from 0.7 to 1.0 mm. Second, brick pellets were treated in our laboratory under the following alkali conditions: 10 g of Bangui brick reacted in 40 mL of a diluted NaOH solution (0.6 mol.L$^{-1}$) at room temperature for one night under slow shaking at a speed of 120 rpm. This procedure was afterwards followed by a fixed-temperature increase in the mixture at 90 °C for a constant reaction time of six days. Finally,

the recovered grains were rinsed several times with MilliQ water and dried at 90 °C for 24 h.

### 2.2. Chemicals

All chemicals employed in the experiments were analytical grades. Sodium hydroxide, sodium nitrate and hydrochloride acid were supplied by DISLAB (Paris, France).

### 2.3. ICP-AES Analyses

Recovered solutions were analyzed for element contents using ICP-OES (Inductively Coupled Plasma—Optic Emission Spectrometry; model 5110 VDV, Agilent Technologies, "Agilent Careers France" based in Paris).

### 2.4. Electron Microscopy Analysis

Micrographies of representative specimens of the brick before and after chemical treatment were recorded by using an environmental scanning electron microscope (ESEM, Quanta 200 FEI). Elemental analysis was performed using ESEM/EDS (ESEM, model: QUANTA–200–FEI, equipped with an Energy Dispersive X-Ray Spectrometer EDS X flash 3001 and monitored by QUANTAX EDS for SEM, Bruker. EDS measurements were carried out at 20 kV at low vacuum (1.00 Torr) and the maximum pulse throughput was 20 kcps. Different surface areas ranging from 0.12 to 0.50 mm$^2$ were targeted on alkali-brick grains and examined by ESEM/EDS. For that, a narrow beam scanned selected areas of brick pellets for chemical analysis. Atomic quantifications and mathematical treatments were undertaken using QUANTA-400 software in order to determine the averaged elemental composition of the surface brick and to detect chemical/elemental variabilities.

### 2.5. $^1$H MAS NMR Analysis

$^1$H MAS-NMR spectra were recorded at 800 MHz, respectively, on a Bruker 18.8 T spectrometer equipped with a Bruker 3.2 mm CP-MAS probe operating at a $\nu_{rot}$ of 20 kHz. $^1$H MAS-NMR experiments were recorded with a $\pi/2$ pulse length of 3.5 µs, 128 transients and a 5 s rd using the DEPTH sequence in order to suppress the signal coming from the measurement probe. $^1$H chemical shifts were referenced as 0 ppm to TMS.

### 2.6. Surface Electrochemical Study

As described previously [36], salt-addition method was used in this work in order to assess whether modified brick had either a negative or positive surface charge in an aqueous medium. Briefly, 1 g of this brick (grains diameter: 0.7–1.0 mm) was placed in each of 10 "80-mL" beakers containing 19.5 mL of Mill-Q water. The pH of the suspensions was adjusted with HCl ($10^{-2}$ mol.L$^{-1}$) or NaOH ($10^{-2}$ mol.L$^{-1}$) to span the expected pH$_{PZC}$ value (i.e., in the pH range: 5.8–6.1). A 0.5 mL volume of a 0.1 mol.L$^{-1}$ NaNO$_3$ solution (as an inert electrolyte) was added to these suspensions. All these mixtures were equilibrated for 1 night, by shaking gently at a constant speed of 120 rpm using a mechanical shaker (Model: IKA Labortechnik KS 250 basic). The equilibrium pH was recorded and designated pH$_{0.0025M}$ (0.0025 M is the final NaNO$_3$ concentration in the medium). Afterwards, a 0.5 mL volume of a 2 M NaNO$_3$ solution was added to the previous mixtures and shaken for a few minutes. The pH was recorded and designated pH$_{0.0525}$ (0.0525 M is the final NaNO$_3$ concentration in the resulting suspension). For each beaker, $\Delta$pH = pH$_{0.0525}$ − pH$_{0.0025}$ was calculated, and $\Delta$pH values were plotted against pH$_{0.0025M}$ in order to evaluate the point where $\Delta$pH = 0 is the point of zero charge, PZC.

## 3. Results

### 3.1. ESEM/EDS Analysis

The brick composite used here was fabricated by surface deposition of zeolites in a support matrix (quartz). This was done by initially producing geo-polymeric gels (from metakaolinite). These gels, which impregnated quartz grains, were progressively crys-

tallized into zeolitic particles dispersed onto $SiO_2$ surfaces. The resulting material then constituted an "adsorptive" membrane in which deposited zeolites were appropriate adsorbents having hydroxyl functional groups capable of interacting favourably with metal ions.

This paragraph had been devoted first to study microscopically surface chemistry/mineralogy aspects of NaOH-activated brick, and second to analyse quantitatively surface minerals constituting the surface composite membrane [14].

The ESEM micrograph of alkali brick displays cubic and spherical specimens (Figure 1). Their sizes varied from 15 μm to 30 μm for cubic specimens and from 6 μm to 15 μm for spherical specimens. Cubic crystals were found to be comparable with those observed previously for the A-type zeolite [37,38]. As for spherical shape crystals, they were found to be morphologically similar to those reported in the literature for zeolite NaP [39–42]. A typical EDS spectrum of cubic or spherical particles is displayed in Figure 2.

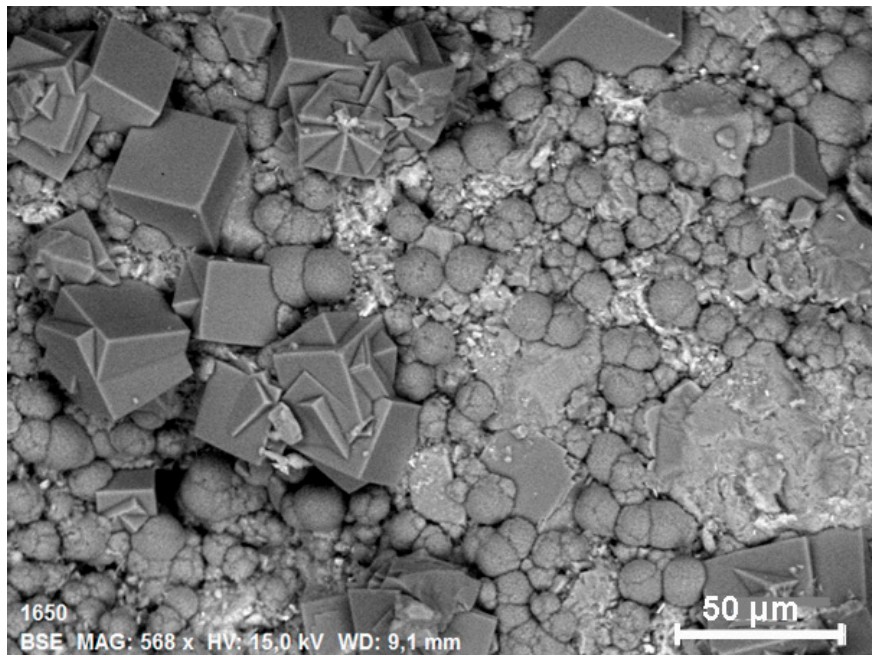

**Figure 1.** ESEM image of composite particles showing aggregated cubic and spherical shapes.

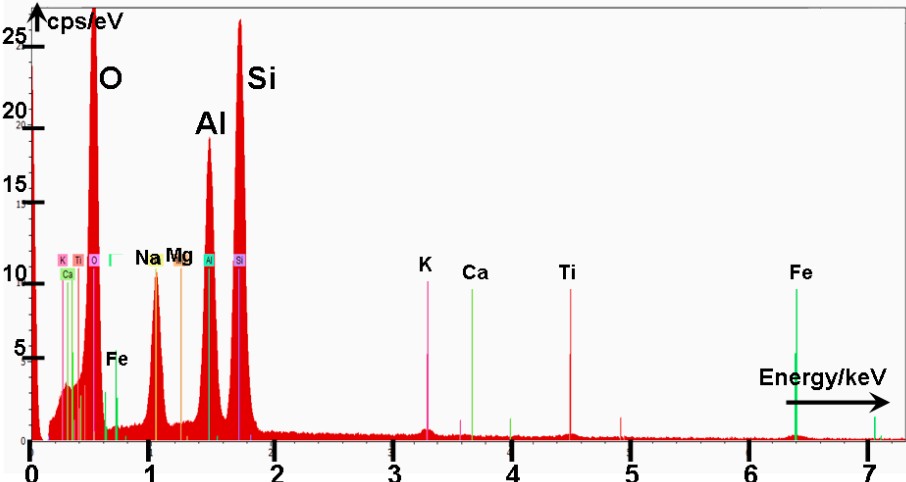

**Figure 2.** Typical energy dispersive X-rays spectrum of zeolitic particles in brick composite.

Quantitative ESEM/EDS analysis indicated that the elemental composition of zeolitic specimens corresponded well enough to those of low-silica zeolites with atomic ratios Si/Al $\approx 1$. As a whole, we found: $14.77 \pm 2.67$ atomic % for sodium; $14.97 \pm 2.05$ atomic % for aluminium; and $16.12 \pm 1.46$ atomic % for silicon. The calculated atomic ratio Si/Al was then equal to about 1.08 (or Al/Si = 0.926). The presence of high amounts of sodium on the surfaces of cubic and spherical particles confirmed the presence of sodic zeolites.

From elemental ESEM/EDS analysis data obtained for brick-derived zeolites, an averaged chemical formulae was established: $Na_{0.93}(H_2O)_z Al_{0.93} Si_{1.0} O_{3.86}$. This formulae was found to be closer enough to that reported in the literature for LTA zeolite ($Na_{1.0}(H_2O)_{0.4} Al_{1.0} Si_{1.0} O_{4.0}$) than that proposed for NaP zeolite ($Na_{0.6}(H_2O)_{1.2} Al_{0.6} Si_{1.0} O_{3.2}$) [43], suggesting the predominance of LTA crystals on alkali-brick surfaces.

In previous works, XRD patterns of alkali-brick powder confirmed the presence of zeolites LTA and NaP in addition to those ascribed to quartz, illite and rutile [12]. Briefly, we detected in the diffractogram the following '2θ' reflection angles (the Miller indices, hkl, are given in the parenthesis): *quartz* 20.9° (100); 26.6° (011); 36.5° (110); 39.5° (102); 40.3° (111); 42.4° (200); 45.8° (201); and 50.1° (112) [ICSD Collection Code: 89276]; *illite* 8.8° (001), 17.9° (004), 19.8° (021), and 34.3° (034) [ICDD (International Centre for Diffraction data): 00-009-0343]; *rutile*: 27.4° (110) and 36.1° (101) [ICSD Collection Code: 168140]; *LTA* 7.2°(200), 10.2°(220), 12.5° (222) and 21.7° (600 and 442) [43]; and *NaP* 12.5° (101 and 110), 17.7° (200 and 002), 21.7°(211, 112 and 121), 28.1°(310, 301, and 103), 33.4° (132, 123, 231, 213, 312, and 321) and 46.1° (134) [43]. It is worth noting that crystallographic peak intensities revealed higher amounts of LTA crystals than NaP crystals, in agreement with quantitative ESEM/EDS surface analyses.

The spatial distribution of the framework elements Al, Fe, Na and Si is displayed in Figure 3A. The ESEM/EDS mapping procedure gives the color overlay shown in this figure, where the elemental distributions for Al, Fe, Na and Si are represented in red, green, yellow, and blue, respectively. Element distribution images indicate a positive correlation between Al, Si and Na (Figure 3B) due to the presence of sodic alumino-silicates as Na-zeolites. Conversely, there is a negative correlation between sodium and silicon in Si-rich zones (composed of quartz crystals). Note, as well, the presence of micro-specimens of $TiO_2$ (rutile) in the elemental distribution for titanium, see Figure 3B.

### 3.2. Protonation of Brick Surfaces in Water

From electro-kinetic measurements (see below), it was demonstrated that brick-composite surfaces were hydroxylated with the formation of hydroxyl groups which behaved amphoterically. This led to noticeable changes in charge/potential on composite surface and cation/anion speciation in solution composition with pH. In what follows, it was assumed that the acid–base properties of alkali brick resulted from zeolitic brick frameworks with single hydroxyl groups. These sites could generate pH-dependent charges at the surface of the material by proton transfers in water. In this article, the functional surface group was expressed as ">S–OH" and its acid-base behaviour was given by:

$$>S\text{-}OH_2^+ \leftrightarrow >S\text{-}OH + H^+_{(aq.)} \tag{1}$$

$$>S\text{-}OH \leftrightarrow >S\text{-}O^- + H^+_{(aq.)} \tag{2}$$

where ">S–O⁻" is an active surface functional group including the silanol (>Si–O⁻) and aluminol (>Al–O⁻) sites; and $H^+_{(aq.)}$ represents a hydroxonium ion in the aqueous phase. Aqueous surface complexation models which were previously used for heterogeneous protonation equilibria on oxides and silicates [44] were applied here to the studied system.

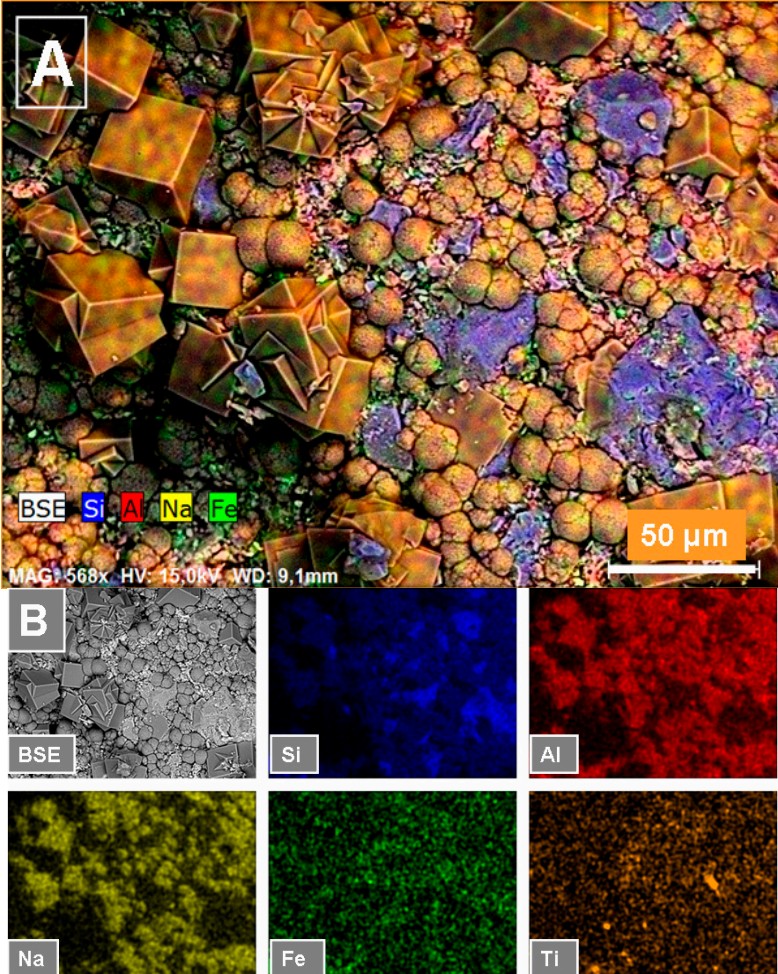

**Figure 3.** Reconstituted ESEM/EDS mapping image of brick composite (**A**) and the spatial distribution of the framework elements: Al, Fe, Na, Si and Ti (**B**).

Sverjensky-Sahais'works on surface protonation had drawn attention to the importance of the pH at the point of zero charge ($pH_{PZC}$) by expressing it in terms of the average bond strength per Angstrom for the solid and the inverse of the dielectric constant of this material [44,45]. With this in mind, in order to predict single-site surface-protonation equilibrium constants for alkali-activated brick in aqueous solutions, we preliminarily estimated Pauling bond strengthsfor Si–$O_i$H, Al−$O_i$H and Si–($O_i$H)–Al in zeolitic brick frameworks (in Section 3.2.1) and the global dielectric constant of the surface composite (in Section 3.2.2).

### 3.2.1. Pauling Bond Strengths in Zeolitic Brick Frameworks

For the NaA zeolite, Fischer and his co-workers [46] showed that the coordination tetrahedra within the framework are rather regular with T-O distances varying from 1.599 to 1.611 Å for the $SiO_4$ tetrahedron and from 1.719 to 1.737 Å for the $AlO_4$ tetrahedron. These studies further revealed that the NaA structure contains $Si^{4+}$ and $Al^{3+}$ cations bound to three different crystallographic oxygens (by considering: $O_i = O_1$, $O_2$ and $O_3$), resulting in three framework distances for each cation (see Table 1).

As for the NaP zeolite, Albert and his co-workers [47] revealed that the NaP structure contains $Si^{4+}$ and $Al^{3+}$ cations bound to four different crystallographic oxygens (by taking: $O_i = O_1$, $O_2$, $O_3$ and $O_4$), leading to four inter-atomic distances for Si and Al noted: Si−$O_i$ and Al−$O_i$ (see Table 1).

Moreover, for the crystalline structure of protonic brick zeolite the different distances ($r_{M-OH}$) were evaluated from the values of the cation-oxygen bond lengths around $Si^{4+}$ and $Al^{3+}$ ions [46,47] and by considering the O$-$H bond length in ice, 1.01 Å [48]:

$$r_{M-Oi-H}= r_{M-Oi} + r_{Oi-H}(ice) \tag{3}$$

with M = Si or Al and i = 1, 2, 3, (and 4 in addition for NaP)

On the other hand, for crystal structures the Pauling bond strength (noted here: 's') is defined as the cation charge (z) divided by its coordination number (noted here: 'n'), i.e.: s = z/n; And the Pauling bond strength divided by radius ($r_{M-Oi-H}$) calculated from Equation (4) or Equation (5) represents the Pauling bond strength per Angstrom ($s/r_{M-OiH}$). The expressions of $s/r_{M-OiH}$ (with i = 1, 2, 3; and 4 in addition for NaP) relative to the different hydroxyl groups (aluminol and silanol) in either NaA or NaP zeolite are given by:

$$\frac{s}{r_{Si-Oi-H}} = \frac{(+4/4)}{r_{Si-Oi} + r_{Oi-H}(ice)} \tag{4}$$

$$\frac{s}{r_{Al-Oi-H}} = \frac{(+3/4)}{r_{Al-Oi} + r_{Oi-H}(ice)} \tag{5}$$

The $s/r_{Si-OiH}$ and $s/r_{Al-OiH}$ values for the zeolitic silanols and aluminols bound to different crystallographic oxygens ($O_i$) are reported in Table 1.

**Table 1.** Pauling bond strengths per Angstrom ($s/r_{M-OH}$) calculated for the different crystallographic distances, M$-$OH (i.e., Si$-$O$_i$H, Al$-$O$_i$H and Si$-$(O$_i$H)$-$Al) of zeolites NaA and NaP.

| NaA | Si$-$O$_{(i)}$ | Al$-$O$_{(i)}$ | $r_{Si-O(i)H}$ | $r_{Al-O(i)H}$ | $r_{Al-(O(i)H)-Si}$ | $s/r_{Si-O(i)H}$ | $s/r_{Al-O(i)H}$ | $s/r_{Al-(O(i)H)-Si}$ |
|---|---|---|---|---|---|---|---|---|
| O$_{(1)}$ | 1.5991 | 1.7189 | 2.6091 | 2.7289 | 2.6690 | 0.3833 | 0.2748 | 0.3290 |
| O$_{(2)}$ | 1.6100 | 1.7240 | 2.6200 | 2.7340 | 2.6770 | 0.3817 | 0.2743 | 0.3280 |
| O$_{(3)}$ | 1.6109 | 1.7371 | 2.6209 | 2.7471 | 2.6840 | 0.3816 | 0.2730 | 0.3273 |
| **NaP** | **Si$-$O$_{(i)}$** | **Al$-$O$_{(i)}$** | **$r_{Si-O(i)H}$** | **$r_{Al-O(i)H}$** | **$r_{Al-(O(i)H)-Si}$** | **$s/r_{Si-O(i)H}$** | **$s/r_{Al-O(i)H}$** | **$s/r_{Al-(O(i)H)-Si}$** |
| O$_{(1)}$ | 1.5920 | 1.7810 | 2.6020 | 2.7910 | 2.6965 | 0.3843 | 0.2687 | 0.3265 |
| O$_{(2)}$ | 1.6240 | 1.7390 | 2.6340 | 2.7490 | 2.6915 | 0.3796 | 0.2728 | 0.3262 |
| O$_{(3)}$ | 1.6620 | 1.6550 | 2.6720 | 2.6650 | 2.6685 | 0.3743 | 0.2814 | 0.3278 |
| O$_{(4)}$ | 1.5910 | 1.7490 | 2.6010 | 2.7590 | 2.6800 | 0.3845 | 0.2718 | 0.3282 |

Besides, by assuming the existence of bridging Si$-$(O$_i$H)$-$Al sites generated by the presence of aluminium inside the silicate framework and the balancing proton, the values of $s/r_{M-OH}$ can be approximated from the relationship:

$$\frac{s}{r_{M-O-H}} = \left[\frac{(+4/4)}{r_{Si-OiH}} + \frac{(+3/4)}{r_{Al-OiH}}\right]/2 \tag{6}$$

The calculated $s/r_{M-OH}$ values for the different bridging Si$-$(O$_i$H)$-$Al sites are listed in Table 1. Finally, an average value of $<s/r_{M-OH}>$ for the global zeolitic structures of alkali brick could be determined from the equation [44]:

$$< \frac{s}{r_{M-O-H}} >= \Sigma_i \left\{\left[\frac{s}{r_{Si-OiH}} + \frac{s}{r_{Al-OiH}}\right]/2\right\}/N_i \tag{7}$$

where $N_i$ represents the numbers of sites or different structural oxygens. We found: $<s/r_{M-OH}>$ = 0.32811 Å$^{-1}$ and 0.32719 Å$^{-1}$ for bridging Si$-$(O$_i$H)$-$Al sites of NaA and NaP zeolites, respectively.

### 3.2.2. Global Dielectric Constant of Surface Alkali-Brick

The estimation of the global dielectric constant of alkali brick preliminarily necessitated the elemental and mineralogical surface composition of this composite. For that purpose, quantitative micro-analytical studies were performed on alkali-brick samples by using

ESEM/EDS. From the reconstituted ESEM/EDS mapping image shown in Figure 3A, one can notice that roughly only two types of elements combinations predominate at alkali-brick surfaces, namely: Si-O and Al-Si-Na which correspond to quartz and zeolites, respectively. With this in mind, we afterwards attempted to evaluate the averaged molecular fractions of these two typical combinations at the brick surface. For that, quantitative ESEM/EDS analysis was performed on at least 35 large 'circle' or 'ellipse' regions of alkali-brick surfaces with diameters ranging from ~200 μm to ~500 μm, as shown in Figure 4.

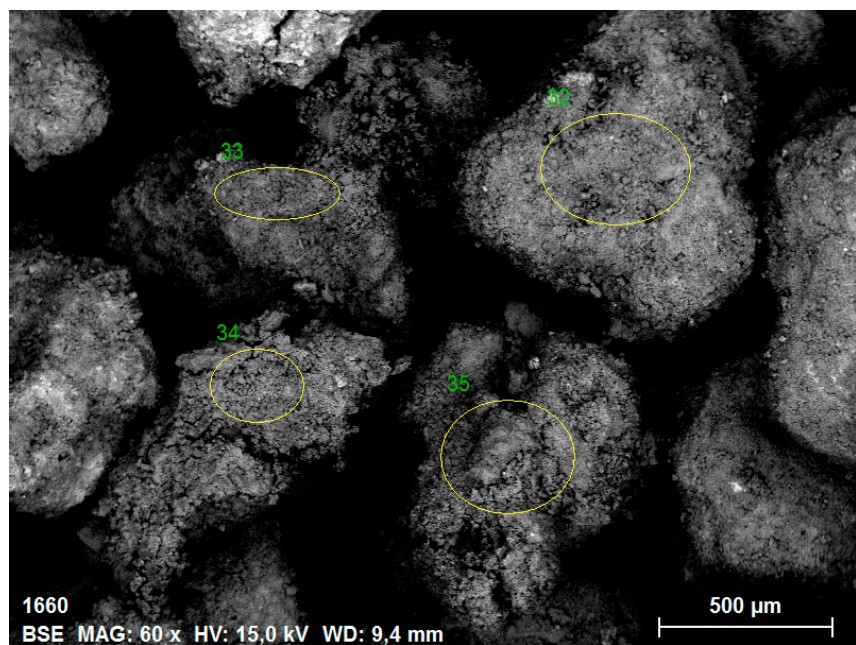

**Figure 4.** Typical 'circle' or 'ellipse' regions of composite surfaces targeted for quantitative ESEM/EDS analysis.

Global ESEM/EDS data indicated that the atomic percentages of Al and Si atoms on alkali-brick were in the following % ranges: 59.65–62.25% for silicon and 9.64–11.31% for aluminium. Taking into account the chemical formulae of quartz and alkali-brick zeolites, it had been possible to estimate the atomic percentage of Si atoms in each of the two mineral forms present at composite surfaces, i.e.: quartz and zeolites. We found: (16.64)–(20.39) atomic % for "zeolite" silicon and (79.61)–(83.36) atomic % for "quartz" silicon. From the volumetric and molecular masses of quartz ($\rho_{quartz}$ = 2.65 g.mL$^{-1}$ [49] and $M_{quartz}$ = 60.09 g.mol$^{-1}$) and zeolite($\rho_{zeolite}$≈ 2.83 g.mL$^{-1}$ [50] and $M_{zeolite}$ ≈ 145.32 g.mol$^{-1}$), it had been possible to approximate the volumetric% of quartz and zeolites at the surface of the alkali-brick composite. We found: (30.72)–(36.71) volumetric % for zeolite and (63.29)–(69.28) volumetric % for quartz. These volumetric % values will be used in what follows for evaluating the approximate dielectric constant of alkali brick. It is worth noting that the dielectric constant estimated in this work would correspond more to that if the constant was really measured at the solid surface.

To summarize, one could assimilate alkali brick to a simple diphase composite of sodic zeolite and quartz with volumetric surface percentages ranging from 30.72% to 36.71% and 63.29% to 69.28%, respectively. These volumetric % values will be used in what follows for evaluating the approximate dielectric constant of surface alkali-brick. It is worth noting that the dielectric constant estimated in this work would correspond more to that if the constant was really measured at the solid surface.

Lichtenecker's equationfor thedielectricfunction of a two-phase composite was suggested in1926 [51,52]:

$$\varepsilon_{dpc}^{\alpha} = V_h.\varepsilon_h^{\alpha} + V_l.\varepsilon_l^{\alpha} \tag{8}$$

where $\varepsilon_{dpc}$ represents the effective dielectric constant of the diphase composite (dpc); $\varepsilon_h$ and $\varepsilon_l$ are the relative dielectric constants of the high-dielectric phase and low-dielectric phase; and $V_h$ and $V_l$ are the volume fractions of the high-dielectric phase and low-dielectric phase, respectively (with $V_h + V_l = 1$). The volume fraction ($V_h$) of the high-dielectric phase (quartz) ranged from 0.6329 to 0.6928 and the volume fraction ($V_l$) of the low-dielectric phase (brick zeolites) ranged from 0.3072 to 0.3671. As for the relative dielectric constants ($\varepsilon_h$ and $\varepsilon_l$) of the high-dielectric phase (quartz) and low-dielectric phase (brick zeolites), we took the following values cited in the literature: $\varepsilon_h = 4.578$ [44] and $\varepsilon_l = 1.62$ [53,54].

In Equation (8), the different types of mixing rules are characterized by the $\alpha$ parameter. For a serial mixing rule (i.e., when $\alpha = -1$), Equation (8) becomes:

$$\frac{1}{\varepsilon_{dpc}} = \frac{V_h}{\varepsilon_h} + \frac{V_l}{\varepsilon_l} \tag{9}$$

For a parallel mixing rule (i.e., when $\alpha = +1$), Equation (8) can be written simply as:

$$\varepsilon_{dpc} = V_h.\varepsilon_h + V_l.\varepsilon_l \tag{10}$$

And for an intermediate (parallel and serial) mixing rule (i.e., when $\alpha \rightarrow 0$), which is also called "logarithmic mixing rule", the dielectric constant of the diphase composite is given by [51,52]:

$$\log \varepsilon_{dpc} = V_h.\log \varepsilon_h + V_l.\log \varepsilon_l \tag{11}$$

In alkali brick, the low-dielectric phase corresponds to zeolitic particles; and these latter are associated with quartz grains which are considered here as the high-dielectric phase in the diphase brick composite. The volume fractions of quartz and zeolites at the brick surface were evaluated by ESEM/EDS analysis (see above). As for $\varepsilon_h$ and $\varepsilon_l$ values, we took the dielectric constants of quartz and LTA zeolite which were both reported previously [44,53,54]. Considering the volume fractions and dielectric constants of quartz and zeolite, Equations (9)–(11) were used for determining the global dielectric constant of the diphase composite: $\varepsilon_{dpc} = \varepsilon_{\text{Alkali-brick}}$ (see Table 2). Averaged $\varepsilon_{\text{Alkali-brick}}$ values are given in Table 2.

**Table 2.** Dielectric constant of the brick composite (considered here as a diphase composite) calculated from Bruggeman's, Lichterecker 's and Hashin-Shtrikman's equations. *<$\varepsilon_{alkali\text{-}brick}$> represents the averaged dielectric constant of alkali brick.*

| Volumic % of Zeolites on Alkali Brick | Dielectric Constant Measured at Alkali-Brick Surfaces ($\varepsilon_{\text{alkali-brick}}$) | | | | |
|---|---|---|---|---|---|
| | Bruggeman | Hashin-Shtrikman | Lichterecker | | |
| | | | á = +1 | á = −1 | á = 0 |
| 36.71% (max.) | **3.272** | 3.150 | 3.492 | 2.741 | 3.126 |
| 30.72% (min.) | 3.473 | 3.346 | 3.669 | 2.933 | 3.327 |
| <$\varepsilon_{\text{alkali-brick}}$> | 3.373 | 3.248 | 3.581 Average: 3.209 | 2.837 | 3.227 |

Hashin and Shtrikman [55] established different mathematical formulae for assessing dielectric constants. These formulae had to be applied more for macroscopically homogeneous and isotopic composites. Hashin and Shtrikman proposed two equations, one as a lower bound:

$$\varepsilon_{dpc} = \varepsilon_l + \frac{V_h}{\frac{1}{\varepsilon_h - \varepsilon_l} + \frac{V_l}{3\varepsilon_l}} \tag{12}$$

And another one as an upper bound:

$$\varepsilon_{dpc} = \varepsilon_h + \frac{V_l}{\frac{1}{\varepsilon_l - \varepsilon_h} + \frac{V_h}{3\varepsilon_h}} \qquad (13)$$

From Equations (12) and (13), the dielectric constant for brick composite was calculated, averaged and listed in Table 2.

Bruggeman employed rather a model of regularly arranged spherical particles [56]. This model was applicable to composite in which both phases have similar morphologies and are distributed randomly through the whole system. Bruggeman symmetrical medium equation was also established in a more general form for oriented ellipsoids [57]:

$$V_h \cdot \frac{\varepsilon_h - \varepsilon_{dpc}}{\varepsilon_h + A.\varepsilon_{dpc}} = (-V_l) \cdot \frac{\varepsilon_l - \varepsilon_{dpc}}{\varepsilon_l + A.\varepsilon_{dpc}} \qquad (14)$$

The A parameter is defined as:

$$A = \frac{1 - V_C}{V_C} \qquad (15)$$

where $V_C$ represents the critical volume fraction of the high dielectric constant phase (for spheres: $V_C = 1/3$). From Equation (14), the dielectric constant of the composite was determined (Table 2).

Overall, as seen in Table 2 the averaged dielectric constants of the composite calculated from Bruggeman's, Lichterecker 's and Hashin-Shtrikman's equations were similar enough each other.

### 3.2.3. Surface-Protonation Equilibria at Composite Surfaces

The constant capacitance model (CCM), the double diffuse layer model (DLM), and the triple layer model (TLM) have been the most frequently used for understanding mineral-water interactions. In the present works, we applied these models to predict surface-pronation equilibrium constants for hydroxyl groups at the surface of the adsorptive material (zeolites NaA and NaP) of the composite membrane.

From single-site models of surface protonation including CCM, DLM and TLM, some authors [44] established mathematical expressions which permitted to predict single-site surface-protonation constant for the zero-point of charge equilibrium (see Table 3):

$$S-OH_2^+ \leftrightarrow >S-O^- + 2H^+_{(aq.)} \qquad (16)$$

**Table 3.** Mathematical expressions of thermodynamic constant ($pK_{PZC}$) for the zero point of charge equilibrium determined from theoretical CCM, DLM and TLM models.

| **Thermodynamic Constant ($pK_{PZC}$) for the Zero Point of Charge Equilibrium) [44]:** $S-OH_2^+ \leftrightarrow >S-O^- + 2H^+_{(aq.)}$ |
|---|
| **CCM:** $pK_{PZC} = 2pH_{PZC} = 22.86(1/\text{å}) - 67.44(s/r_{M-OH}) + 26.76$ |
| **DLM:** $pK_{PZC} = 2pH_{PZC} = 22.14(1/\text{å}) - 66.98(s/r_{M-OH}) + 26.78$ |
| **TLM:** $pK_{PZC} = 2pH_{PZC} = 42.2316(1/\text{å}) - 85.8296(s/r_{M-OH}) + 29.3732$ |

Note that Equation (16) corresponds well to the sum of the two surface- protonation reactions mentioned above: Equation (1) + Equation (2). In this paragraph, from calculated Pauling bond strength values for $Si-O_iH$, $Al-O_iH$ and $Si-(O_iH)-Al$ in surface-brick frameworks and surface dielectric-constant value (see Sections 3.2.1 and 3.2.2), we attempted to assess the zero-point of charge equilibrium constant and surface-protonation equilibrium constants.

As a first approach, the diffuse double Layer Model (DLM) was initially applied here for assessing the pH values at the point of zero charge ($pH_{PZC}$) for different Brønsted acids that might exist in the crystalline structure of protonic brick zeolites.

$pH_{PZC}$ values were determined from the Diffuse double Layer Model (DLM). For that purpose, we used the different values of Pauling bond strengths per Angstrom which were reported in Table 1 for M−OH distances (i.e., $Si−O_iH$, $Al−O_iH$ and $Si−(O_iH)−Al$) and by taking a dielectric constant value of surface alkali-brick averaged from 'Table 2' data, i.e.: $<\varepsilon_{alkali-brick}> = 3.291$ (Table 4).

**Table 4.** Theoretical predictions of $pH_{PZC}$ values for different Brønsted acids present in the crystalline structure of protonic brick zeolites by applying the Diffuse double Layer Model.

| NaA | $s/r_{Si−O(i)H}$ | $pH_{PZC}$ (SiOH) | $s/r_{Al−O(i)H}$ | $pH_{PZC}$ (AlOH) | $s/r_{Al−(O(i)H)−Si}$ | $pH_{PZC}$ (AlOHSi) |
|---|---|---|---|---|---|---|
| $O_{(1)}$ | 0.38327 | 4.31 | 0.2748 | 7.94 | 0.3291 | 5.73 |
| $O_{(2)}$ | 0.38168 | 4.36 | 0.2743 | 7.96 | 0.3280 | 5.77 |
| $O_{(3)}$ | 0.38155 | 4.37 | 0.2730 | 8.00 | 0.3273 | 5.79 |
| NaP | $s/r_{Si−O(i)H}$ | $pH_{PZC}$ (SiOH) | $s/r_{Al−O(i)H}$ | $pH_{PZC}$ (AlOH) | $s/r_{Al−(O(i)H)−Si}$ | $pH_{PZC}$ (AlOHSi) |
| $O_{(1)}$ | 0.38432 | 4.27 | 0.2687 | 8.14 | 0.3265 | 5.82 |
| $O_{(2)}$ | 0.37965 | 4.43 | 0.2728 | 8.01 | 0.3262 | 5.83 |
| $O_{(3)}$ | 0.37425 | 4.61 | 0.2814 | 7.72 | 0.3278 | 5.77 |
| $O_{(4)}$ | 0.38447 | 4.27 | 0.2718 | 8.04 | 0.3282 | 5.76 |

On the other hand, the $pH_{PZC}$ of the composite was determined experimentally by using the salt-addition method [36], see Figure 5.

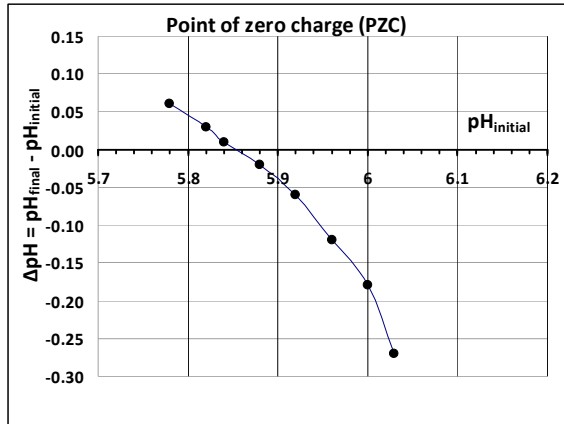

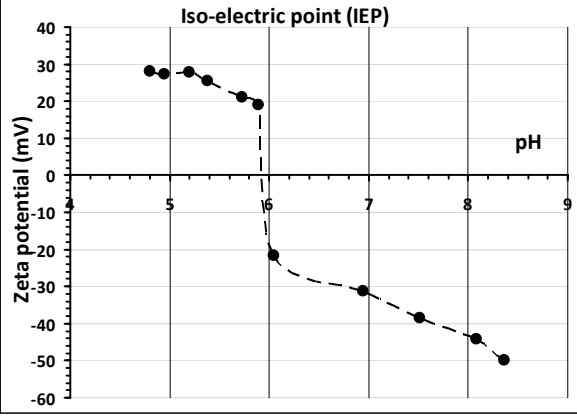

**Figure 5.** "ΔpH" variations versus "initial pH" for the determination of the pH at the point of zero charge of the composite by means of the salt-addition method. "Zeta potential" versus "suspension pH" for the determination of the pH at the iso-electric point of the composite by zetametry.

We found $pH_{PZC}$ = 5.86. This experimental $pH_{PZC}$ value was compared with calculated $pH_{PZC}$ values and found to be closer to the theoretical $pH_{PZC}$ ones predicted for bridging $Si-(O_iH)-Al$ sites (5.73 < $pH_{PZC}$ < 5.83; see Table 4) than those for $Al-O_iH$ sites (7.72 < $pH_{PZC}$ < 8.14) and for $Si-O_iH$ sites (4.27 < $pH_{PZC}$ < 4.61). This suggested that Brønsted acidity in the composite was intimately related to the occurrence of $Si-(O_iH)-Al$ bonding in the bulk crystal structure.

Taking into account the different calculated values of $s/r_{M-OH}$ listed in Table 1 and the averaged dielectric constant value of surface alkali-brick ($<\varepsilon_{alkali-brick}>$ = 3.291), the mathematical expressions given in Table 3 permitted calculation of values of $pH_{PZC}$ for CCM, DLM and TLM models. Predicted and experimental $pH_{PZC}$ values were afterwards plotted against the Pauling bond strength per Angstrom, $s/r_{MOH}$ (Figure 6).

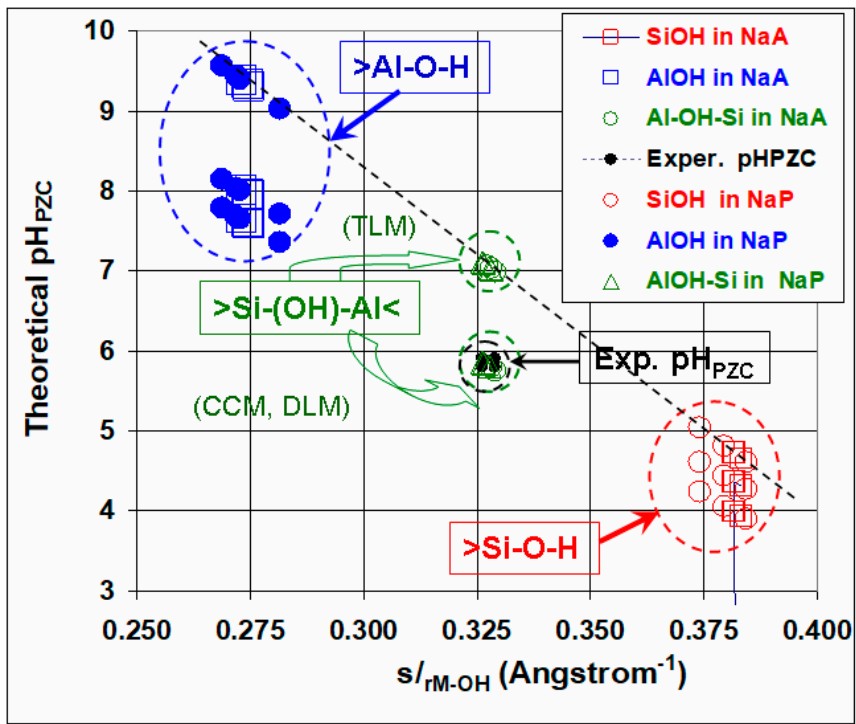

**Figure 6.** Theoretical pH versus Pauling bond strength per Angstrom ($s/r_{M-OH}$) calculated for the different distances, $M-OH$ (i.e., $Si-O_iH$, $Al-O_iH$ and $Si-(O_iH)-Al$) inside protonic brick zeolites.

In this figure, the points corresponding to the $pH_{PZC}$ values which were determined from TLM model are situated along the dashed line. As can be seen in Figure 6, the $pH_{PZC}$ values for the different bridging $Si-(O_iH)-Al$ sites__ which were determined from crystallographic data__ were found to be similar enough to that obtained experimentally (particularly, when applying CCM and DLM models). Conversely, the $pH_{PZC}$ values for $Si-O_iH$ and $Al-O_iH$ sites, which were evaluated from CCM, DLM and TLM models, were relatively far from the experimental one ($pH_{PZC}$ = 5.86).

From the different values of $\varepsilon_{alkali-brick}$ determined from Bruggeman's, Lichterecker 's and Hashin-Shtrikman's equations (see Table 2) and Pauling bond strengths per Angstrom calculated for bridging $Si-(O_iH)-Al$ sites of the modified brick (see Table 1), both $pK_{PZC}$ ($K_{PZC}$: thermodynamic constant for the zero-point of charge equilibrium) and $pH_{PZC}$ were estimated from CCM, DLM and TLM models (Table 5).

**Table 5.** Theoretical predictions of the logarithm of the thermodynamic constant for the zero-point of charge equilibrium ($pK_{PZC}$) and $pH_{PZC}$ for bridging Si$-$(O$_i$H)$-$Al sites of the brick composite from CCM; DLM and TLM models [*Dielectric constants of alkali brick determined from Bruggeman's, Lichterecker 's and Hashin- Shtrikman's equations (see* Table 2) *and Pauling bond strengths per Angstrom evaluated for bridging Si$-$(O$_i$H)$-$Al sites (see* Table 1), *were employed in the calculation*].

| Dielectric Constant ($\varepsilon_{alkali\ brick}$) | | $pK_{PZC}$ | | | $pH_{PZC}$ | | |
|---|---|---|---|---|---|---|---|
| | | **CCM** | **DLM** | **TLM** | **CCM** | **DLM** | **TLM** |
| **Bruggeman** | **3.373** ± **0.100** | **11.78** ± **0.09** | **11.41** ± **0.09** | **13.79** ± **0.12** | 5.89 ± 0.05 | 5.70 ± 0.05 | 6.89 ± 0.06 |
| **Hashin-Shtrikman** | 3.248 ± 0.098 | 12.04 ± 0.09 | 11.66 ± 0.09 | 14.26 ± 0.12 | 6.02 ± 0.05 | 5.83 ± 0.05 | 7.13 ± 0.06 |
| **Lichterecker (serial/parallel Form)** | 3.581 ± 0.089 | 11.38 ± 0.09 | 11.02 ± 0.09 | 13.05 ± 0.12 | 5.69 ± 0.05 | 5.51 ± 0.05 | 6.53 ± 0.06 |
| **Lichterecker (log. Form)** | 3.227 ± 0.100 | 12.08 ± 0.09 | 11.70 ± 0.09 | 14.35 ± 0.12 | 6.04 ± 0.05 | 5.85 ± 0.05 | 7.18 ± 0.06 |

By examining this Table, one could notice that the theoretical $pH_{PZC}$ and $pK_{PZC}$ values predicted from the CCM model ($5.69 < pH_{PZC} < 6.04$ and $11.38 < pK_{PZC} < 12.08$) and DLM model ($5.51 < pH_{PZC} < 5.85$ and $11.02 < pK_{PZC} < 11.70$),were well consistent with the experimental $pH_{PZC}$ and $pK_{PZC}$ values ($pH_{PZC} = 5.86$ and $pK_{PZC} = 11.72$); Whereas the theoretical $pH_{PZC}$ and $pK_{PZC}$ values predicted from the TLM model ($6.53 < pH_{PZC} < 7.18$ and $13.05 < pK_{PZC} < 14.35$) were inconsistent with the experimental ones, as also evidenced in Figure 6.

Surface-protonation equilibrium constants ($K_{\alpha1}$ and $K_{\alpha2}$) for reactions (1) and (2) could also be predicted from mathematical expressions deduced from CCM, DDM and TLM models (see Table 6) [44].

**Table 6.** Mathematical expressions of surface-protonation equilibrium constants ($K_{\alpha1}$ and $K_{\alpha2}$) for reactions (1) and (2) determined from theoretical CCM, DLM and TLM models.

| Surface-Protonation Equilibrium Constants ($K_{á1}$ and $K_{á2}$) for Reactions (1) and (2) [44] |
|---|
| *Constant capacitance model (CCM):* |
| $pK_{á1}(CCM) = ll.43(1/å) - [45.32 - 2.3 \log I](s/r_{M-OH}) + 14.18$ |
| $pK_{á2}(CCM) = ll.43(1/å) - [22.13 + 2(.3 \log I](s/r_{M-OH}) + 12.58$ |
| *Diffuse double layer model (DLM):* |
| $pK_{á1}(DLM) = ll.07(1/å) - 48.50(s/r_{M-OH}) + 14.51$ |
| $pK_{á2}(DLM) = ll.07(1/å) - 18.49(s/r_{M-OH}) + 12.27$ |
| *Triple layer model (TLM):* |
| $pK_{á1}(TLM) = 21.1158(1/å) - 49.2608(s/r_{M-OH}) + 12.9181$ |
| $pK_{á2}(TLM) = 21.1158(1/å) \ 36.5688(s/r_{M-OH}) + 16.4551$ |

"I": the ionic strength of the solution in contact with the solid phase.

Surface-protonation equilibrium constants of brick composite were estimated by taking an approximated constant ionic strength of $0.2\ mol.L^{-1}$ and "Pauling bond strength per Angstrom" and "dielectric constant" values of surface alkali-brick averaged from 'Table 2' data: $<s/r_{M-OH}> = 0.3276$ and $<\varepsilon_{alkali-brick}> = 3.291$. From the $pK_{\alpha1}$ and $pK_{\alpha2}$ values listed in Table 7, we could make the following remarks. The averaged $pK_{\alpha1}$ and $pK_{\alpha2}$ values found with CCM, DLM ($pK_{\alpha1} = 1.99–2.28$ and $pK_{\alpha2} = 9.33–9.58$, respectively) were in the

range of those determined experimentally in a recent work, i.e.:$pK_{\alpha 1} = 2.65 \pm 0.42$ and $pK_{\alpha 2} = 8.90 \pm 0.37$ [58].

**Table 7.** Evaluation/prediction of averaged surface-protonation equilibrium constants for the brick composite from theoretical CCM, DLM and TLM models.

| Averaged pK (Alkali Brick) | Surface Equilibrium Constants | | |
|---|---|---|---|
| | **CCM** | **DLM** | **TLM** |
| **$\langle pK_{\acute{a}1} \rangle$** | $2.28 \pm 0.07$ | $1.99 \pm 0.07$ | $3.20 \pm 0.06$ |
| **$\langle pK_{\acute{a}2} \rangle$** | $9.33 \pm 0.03$ | $9.58 \pm 0.03$ | $10.89 \pm 0.05$ |
| **$\langle pK_{n} \rangle$** | $7.05 \pm 0.10$ | $7.58 \pm 0.10$ | $7.69 \pm 0.11$ |

On the other hand, another surface equilibrium was studied:

$$2 >S-OH \leftrightarrow >S-OH_2^+ + >S-O^- \tag{17}$$

This equilibrium is obtained by subtracting reaction (1) to reaction (2). The main interest of reaction (17) is that no $H_3O^+$ ions from the solution are involved in the system and only surface species are taken into account. The surface equilibrium constant for reaction(17) which is often termed $\Delta pK$ (or $pK_n$), can be calculated from the following expression: $\Delta pK = pK_n = pK_{\alpha 2} - pK_{\alpha 1}$, see Table 7. The $pK_n$ values which were obtained from CCM, DLM and TLM models ($pK_n = 7.05$–$7.69$), were found to be consistent enough with that estimated from the experimental $pK_{\alpha 1}$ and $pK_{\alpha 2}$ values, i.e.: $6.25 \pm 0.79$ [58].

### 3.2.4. $^1$H MAS NMR Analysis of Brønsted Acid Sites of Brick Composite

Ammonia/ammonium was widely used in the past as a probe molecule in NMR spectroscopy for characterizing Brønsted acid sites in alumino-silicate frameworks. In this paragraph, we attempted to gain information on the chemical nature of hydroxyl groups in brick composite when acidifying it slightly at $4.4 < pH < 5.0$ (in order to avoid the decomposition of zeolites) and reacting it with ammonium ions.

A suspension of composite grains (diameter: 0.7–1.0 mm) in Milli-Q water was acidified progressively up to reach a medium pH of ~4.6. The $^1$H MAS NMR spectrum of the acidified composite revealed the presence of a sharp peak at $\delta = 4.7$ ppm ascribed to $H_3O^+$ ions bound to brick zeolites, see Figure 7A. This signal was indeed due to hydrogen nuclei in bridging (structural) hydroxyl groups that underwent a rapid exchange between water molecules and hydroxonium ions in zeolitic cages. Such an observation agreed noticeably well with previous studies about the formation of hydroxyl groups in low-silica zeolite with $n_{Si}/n_{Al} = 1$ (during the exchange of $Na^+$ cations by ammonium ions), particularly showing close signals in the chemical shift range of 3.6–4.8 ppm due to Si(OH)Al groups in the $\alpha$ and $\beta$-cages [34]. The resonance at $\delta_{1H} = 4.7$ ppm was also found to be similar enough to that observed in the spectra of X and Y zeolites and ascribed to Si(OH)Al groups in sodalite cages [33,59].

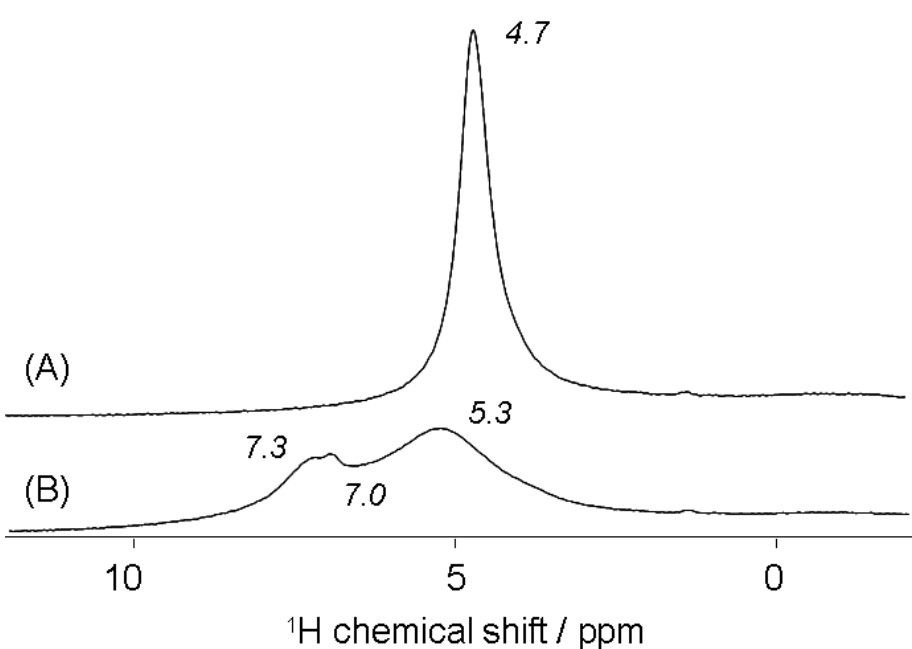

**Figure 7.** $^1$H MAS NMR spectra of the acidified alkali-brick at pH 4.6 before (**A**) and after ammonium treatment (**B**).

Moderately-acidified-composite pellets were subsequently treated with a NH$_4$Cl solution ([NH$_4$Cl] = 0.5 mol.L$^{-1}$). The $^1$H MAS NMR spectrum of the treated composite displayed three new proton resonances (Figure 7B). The peaks at $\delta_1 \cong 7.3$ ppm and $\delta_{1'}$ = 7.0 ppm were assigned both to NH$_4^+$ ions adsorbed onto Bronsted sites of framework zeolites: LTA and NaP (note that the $\delta_1$ and $\delta_{1'}$ peaks were not yet assigned to specific zeolites). Whereas the peak at $\delta_2$ = 5.3 ppm was attributed to ammonium species involved in fast exchanges with NH$_4^+$ ($\delta_1$ and $\delta_{1'}$) and hydroxonium ions in excess. A comparable magnetic phenomenon was already evidenced for other zeolites [60,61].

Upon H$_3$O$^+$/NH$_4^+$ exchange, the resonance of zeolitic $^1$H nuclei was low-field-shifted by $\Delta\delta_1$ = 2.45 ppm (from 4.70 to ~7.15) and $\Delta\delta_2$ = 0.60 ppm (from 4.70 to 5.30), which globally reflected an increasing acid strength of brick protons. Such low-field-shifts induced by ammonium ions were previously observed in $^1$H Solid-state NMR studies of Brønsted acid sites of other zeolites [33,35,60,61] and flame-derived silica/alumina [32] in which silanols with neighbouring aluminium atoms generated bridging AlOHSi groups.

Complementary MAS NMR experiments on the brick composite [one-dimensional $^1$H, $^{27}$Al, and $^{29}$Si NMR and two-dimensional H$-^{27}$Al and $^{29}$Si$-^{27}$Al dipolar correlation (D-HMQC) NMR] are still under way in the lab. Indeed, as suggested previously [62–64] such NMR investigations would be useful to gain more information about Si-OH-Al connections in alumino-silicate frameworks.

## 4. Conclusions

A brick from Central African Republic was treated with sodium hydroxide at 90 °C for 6 days. Surface analysis of treated material showed the predominance of quartz and low-silica zeolites (NaA and NaP), in agreement with XRD results. Quantitative ESEM/EDS studies permitted to determine the averaged mineralogical composition of surface minerals. This mixed material was assimilated to an "adsorptive" membrane, having zeolite particles deposited onto a support matrix (quartz).

Averaged dielectric constant for the surface composite was calculated from various empirical equations proposed in the literature for diphase composites. Surface characteristics of Brønsted acid sites were considered here as being intimately related to the crystalline nature and dielectric surface properties of the mixed material. By applying surface complexation theory, modelling calculations permitted to reveal that: (i) surface

protonation equilibrium constant was much better predicted from CCM and DLM models than from TLM model; and (ii) by considering bridging $Si-(O_iH)-Al$ sites, CCM and DLM models predicted better the thermodynamic constant for the zero-point of charge equilibrium and surface-protonation equilibrium constants than from TLM model. The existence of bridging Brønsted acid sites at acidified composite surfaces interacting with ammonium (as probe ion) was proved by using $^1H$ MAS NMR spectroscopy.

Globally, the theoretical basis developed in the present study was of considerable use in helping to analyse and understand the surface chemistry of low-silica zeolites when dispersed onto a support matrix (quartz) to form an adsorptive composite membrane. Mostly, it highlighted the implication of Brønsted acidic bridging OH groups (Si(OH)Al) on the electro-kinetic behaviour of the studied zeolites-quartz composite during acidification. However, complementary MAS NMR experiments on the brick composite [mainly two-dimensional $^1H-^{27}Al$ and $^{29}Si-^{27}Al$ dipolar correlations (D-HMQC) NMR] would be very useful for obtaining more details of aluminum and silicon environments in acidified-composite frameworks, and particularly, tetrahedral framework Al sites known to give rise to bridging-acid site hydroxyl groups. These NMR results should be published soon in another paper.

Our results would provide a rational basis for further studies on micro-structural changes inside brick-composite membranes with different Si/Al ratios, controlling membrane capacitance/conductance and regulating permeability and selectivity. Numerical (column) simulation of selective heavy metals adsorption/desorption on brick composite with structural change of zeolite membrane with Al/(Al+Si) ratio should further be addressed in our research perspectives by using finite element analysis.

Future research should also focus on studying other types of materials like "ceramic" adsorptive membranes with natural clay materials, since past studies focused mostly on polymeric (organic) adsorptive membranes. Our future research will concern the fabrication of new zeolitic adsorptive membranes by introducing transition metals like Fe and Mn into zeolite frameworks deposited onto quartz matrix. The presence of hetero-atoms in molecular structures of zeolites might be an effective way to improve the adsorption performance of composite membrane towards charged organic/inorganic pollutants.

**Author Contributions:** Conceptualization, A.B., O.A. and M.W.; Formal analysis, A.B., O.A., N.P., G.T., B.R. and L.L.; Investigation, A.B., O.A. and M.W.; Methodology, A.B.; Software, A.B., O.A., G.T. and B.R.; Resources, N.P., G.T., B.R. and L.L.; Validation, A.B., O.A. and M.W.; Visualization, A.B.; Project administration, M.W.; Writing—original draft, A.B.; Writing—review & editing, A.B. All authors have read and agreed to the published version of the manuscript.

**Funding:** This research received no external funding.

**Data Availability Statement:** Not applicable.

**Acknowledgments:** These scientific works were undertaken successfully owing to the cooperation between the University of Lille (France) and the University of Bangui (Central African Republic). ICP-AES and ICP-MS measurements were performed on the Chevreul Institute Platform (U-Lille/CNRS). The Region "Hauts de France" and the French government are warmly acknowledged for the co-funding of these apparatus. The authors gratefully thank David Dumoulin (Chemical Engineer) and Véronique Alaimo (Chemical Technicians) for helping us usefully in certain delicate chemical and analytical/spectroscopic analyses. Scanning electron microscopy studies were undertaken in the laboratory UMR LOG 8187 at the department of Earth Sciences (Villeneuve d'Ascq 59655, France). The authors gratefully thank Sandra Vantalon (Chemical Engineer) for carrying out carefully ESEM/EDS experiments.

**Conflicts of Interest:** The authors declare that they have no known competing financial interests or personal relationships that could have appeared to influence the work reported in this paper.

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
