# Peer review of "Micro-Analytical Study of a Zeolites/Geo-Polymers/Quartz Composite, Dielectric Behaviour and Contribution to Brønsted Sites Affinity"

_ceramics, doi:10.3390/ceramics5040066_

Round 1

Reviewer 1 Report

1.      The abstract should be broadened to give additional quantitative results.

2.      At the end of your abstract, please provide a "take-home" message.

3.      All of the keywords need to use lowercase based on MDPI format.

4.      Please do not use abbreviations in keywords.

5.      Novelty in the current study's is too weak. The past has seen an extensive study of a lot of written material. It is required to provide more details for more explanation about the present novel in the introductory section.

6.      Previous study related needs to explain in the introduction section consisting of their work, their novelty, and their limitations to show the research gaps that intend to be filled in the present study.

7.      Why the present study performing experimental testing? Why not computational simulation or analytical solution? Need more discussion.

8.      Potential study of computational simulation using finite element analysis to study materials performance is needed to mention. It is a vital topic that authors must provide in the introduction and/or discussion section. Additionally, the suggested reverence should be taken to substantiate this explanation as follows: Tresca Stress Evaluation of Metal-on-UHMWPE Total Hip Arthroplasty during Peak Loading from Normal Walking Activity. Mater. Today Proc. 2022, 63, S143–6. https://doi.org/10.1016/j.matpr.2022.02.055

9.      In the materials and methods, the authors need to add additional illustrations as a form of figure that explains the workflow of the present study to make the reader easier to understand rather than only the dominant text as a present form.

10.   Other information about the tool, such as the manufacturer, country, and specifications, should be provided.

11.   The error and tolerance of the experimental tools employed in this investigation are critical details that must be explained in the publication. It would be a valuable discussion because of the differing outcomes in the subsequent study by other researchers.

12.   A comparative assessment with similar previous research is required.

13.   What is the limitation of the present work? Please include it before the conclusion section.

14.   In the conclusion section, discusses future research that is required.

15.   The reference should be enriched with literature from the last five years. Literature published by MDPI is strongly recommended.

16.   In the entire manuscript, the authors occasionally constructed paragraphs with just one or two phrases, which made the explanation difficult to understand. To make their explanation a full paragraph, the authors should expand it. It is advised to use at least three sentences in a paragraph, with the primary sentence coming first and the supporting sentences coming after. See line 73-74.

17.   Due to grammatical problems and linguistic style, the authors should proofread the work. It would be used MDPI English editing service for this concern.

18.   Please review and confirm that the writers followed the MDPI format exactly, edit the current form, and recheck in addition to the other issues that have been mentioned.

Author Response

Villeuve d’Ascq ‘France), September 29th 2022

To Reviewer 1

Dear Madam or Sir,

We have revised the manuscript according Reviewers’ recommendations and suggestions. 

We have made the following corrections and modifications (the added text is typed in red).

(1)The abstract has been broadened with additional data.

(2)A message about “ceramic” adsorptive membranes with natural clay materials has been added at the end of the abstract.

(3)Keywords have been modified.

(4)In keywords, abbreviations have been suppressed.

(5)Novelty is now more explained in the introduction section.

(6)The introduction has been entirely changed with more details and explanations.

(7)In the revised version, we had explained why no electrochemical analysis and simulation had been performed on brick / water suspensions.

(8)Stilling-Brantley‘s works (Refs. 16, 17) have been taken to substantiate difficulties  for simulating acid-base characteristics of mixed oxides like those observed in the present work.

(9)In order to make the reader easier to understand, we have preferred to mention the different objectives of our work in the introduction part.

(10)Details such as manufacturer, country and specifications have been given in the text according to our recent knowledge.

(11)Number of digits has been diminished in table values.

(12)In Sverjensky-Sahai’s works and other works, surface protonation constants were reported only for pure silica and alumina oxides. To our knowledge, no so detailed thermodynamic study was addressed for an “adsorptive” composite membrane like that studied here.

(13)1H MAS NMR results have been added in the text for supporting the existence of bridging Brønsted acid sites at the surface of the adsorptive material deposited onto quartz matrix..

(14)Future research is discussed in the conclusion section.

(15)The reference section has been enriched with more references.

(16)At the beginning of each paragraph, some sentences have been added in the text for better clarification and explanation.

(17)The English has been checked at best.

We gratefully thank you for helping us to improve significantly the scientific content of this manuscript.

Best regards

Dr. A. Boughriet

Reviewer 2 Report

The manuscript describes the alkaline treatment of a brick made by craftsmen in Bangui region and the subsequent study of the surface protonation of the composite. The dielectric constant of the treated brick was calculated from Bruggeman’s, Hashin- Shtrikman’s and Lichterecker ‘s equations by using the volume fraction of quartz and zeolite estimated from the ESEM/EDS analysis. The surface-protonation equilibrium constants of the brick have been calculated from the CCM, DDM and TLM models and compared with the experimental results. The CCM and DLM model showed a better prediction of the constant than the TLM model.

The manuscript is well structured and provides a detailed explanation of the experimental procedures and calculations used to predict the values of the constants. I consider that the theoretical prediction of surface-protonation equilibrium constants is of interest to the material science community and this work will help to better understand the solid surface charge.

Nevertheless, I have some recommendations to improve the manuscript.

1.       I recommend a revision of English grammar and scientific terminology.

2.       I recommend improving the introduction section, including an explanation of what a brick is, and stronger motivation for the study.

3.       Use the same nomenclature, e.g. Zn(II) and Zn2+ in the introduction section, Bronsted and Brönsted, SiO- and >Al-O-.

4.       In the results part, it is claimed that the alkali-treated brick contains quartz, zeolite A and zeolite P. Please, show the XRD pattern of the material or provide any reference where this material was previously analyzed. ESEM microscopy is not an accurate method to identify crystallographic phases.

5.       Please, consider the significant digits of the reported values

Author Response

Villeuve d’Ascq ‘France), September 29th 2022

To Reviewer 2

Dear Madam or Sir,

We have revised the manuscript according Reviewers’ recommendations and suggestions.

We have made the following corrections and modifications (the added text is typed in red).

(1)English grammar has been verified at best.

(2)Introduction has been entirely re-written with more clarification and explanations.  The objectives of this work have been well detailed.

(3)Nomenclature has been checked and Brønsted is now well spelled. SiO- or >AlO- has been changed.

(4)Detailed XRD pattern of the material is now given in section 3.1.

(5)The number of digits has been diminished in the different tables.

We gratefully thank you for helping us to improve significantly the scientific content of this manuscript.

Best regards

Dr. A. Boughriet

Reviewer 3 Report

Ms. Ref. No.   : ceramics-1918541

Title: Micro-Analytical Study of a Zeolites / Geo-Polymers / Quartz 2 Composite, Dielectric Behaviour and Contribution To 3 Bronsted Sites Affinity

Overview and general recommendation:

The paper presents detail on the composite structure, dielectric behavior, and its contribution to the affinity of the Bronsted site. However, there are several things I must subject and validate:

1.      In the paper, a thorough discussion on the intrinsic properties of quartz and composites was presented. It is understandable that the brief purpose of quartz is mostly used as membranes. However, there is no discussion on the relation of the intrinsic properties discussed by the authors with the “membrane” purpose of quartz and composites. It is recommended to add more information on the relation of the intrinsic properties of quartz with the membrane application.

2.      It is suggested to explore and discuss the purpose of understanding the dielectric behavior and its contribution to the affinity of the Bronsted site of composite.

3.      The structure, as well as the composition of the composites, are the main points of this paper to be addressed accordingly. Therefore, rather than assuming the composition of composites based on ESEM-EDS, whether it is zeolite P or A, it would be more precise to use powder XRD measurement and compare the pattern with the database of XRD on IZA or ICDD. Furthermore, it will be advantageous if the materials are also analyzed by 29Si and 27Al NMR to observe the Si-OH-Al connection in the zeolite framework: for example: [DOI: 10.3390/app11156850, 10.1016/j.micromeso.2018.10.007, 10.1016/j.coche.2019.04.002, 10.1021/jacs.1c02361, 10.3390/molecules25010031].

Therefore, the observation of the position of Si, Al, and OH bridge that contribute to the Bronsted site and dielectric properties of the materials can be elucidated and compared to the simulated ones.

4.      Please include some references to your conclusion on the molecule formulas of materials based on EDS results. Bear in mind that it is rather difficult to decide that the composites are dominated by zeolite P based on the formula given by EDS since EDS only gives overall elemental analysis for the composites (e.g. the combination of zeolite P and A, and not in their single face).

5.      Lastly, there were many typo mistakes found in the manuscript. 

Author Response

Villeuve d’Ascq (France), September 29th 2022

To Reviewer 3

Dear Madam or Sir,

We have revised the manuscript according Reviewers’ recommendations and suggestions.

We have made the following corrections and modifications (the added text is typed in red).

(1)A discussion on the adsorptive membrane characteristics of the brick composite (with quartz as support matrix) is now made in the introduction section. Recent “REVIEW” references relative to membrane applications are also cited in the text.

(2)A better understanding of the dielectric behaviour of the composite and its contribution to the affinity of Brønsted sites has been addressed in the manuscript on a thermodynamic base of interfacial water/solid phenomena developed by James and Healy. In addition, to confirm the existence of bridging Brønsted acid sites at zeolite surfaces, 1H MAS NMR spectroscopy had been used successfully and NMR data are now given in the text. However, more detailed NMR investigations are still under way to study two-dimensional 1H—29Si, 1H−27Al and 29Si−27Al dipolar correlations (D-HMQC) NMR.

(3)XRD measurements were previously performed on composite samples. A reference is cited in the text and a brief comment on obtained XRD patterns is made in this new version.

(4)References relative to XRD characteristics of NaA and NaP are now cited in the text. Note that XRD analysis permitted us to evidence the predominance of NaA crystals to NaP crystals.

(5)English grammar has been checked at best.

We gratefully thank you for helping us to improve significantly the scientific content of this manuscript.

Best regards

Dr. A. Boughriet

Villeuve d’Ascq (France), September 29th 2022

To Reviewer 3

Dear Madam or Sir,

We have revised the manuscript according Reviewers’ recommendations and suggestions.

We have made the following corrections and modifications (the added text is typed in red).

(1)A discussion on the adsorptive membrane characteristics of the brick composite (with quartz as support matrix) is now made in the introduction section. Recent “REVIEW” references relative to membrane applications are also cited in the text.

(2)A better understanding of the dielectric behaviour of the composite and its contribution to the affinity of Brønsted sites has been addressed in the manuscript on a thermodynamic base of interfacial water/solid phenomena developed by James and Healy. In addition, to confirm the existence of bridging Brønsted acid sites at zeolite surfaces, 1H MAS NMR spectroscopy had been used successfully and NMR data are now given in the text. However, more detailed NMR investigations are still under way to study two-dimensional 1H—29Si, 1H−27Al and 29Si−27Al dipolar correlations (D-HMQC) NMR.

(3)XRD measurements were previously performed on composite samples. A reference is cited in the text and a brief comment on obtained XRD patterns is made in this new version.

(4)References relative to XRD characteristics of NaA and NaP are now cited in the text. Note that XRD analysis permitted us to evidence the predominance of NaA crystals to NaP crystals.

(5)English grammar has been checked at best.

We gratefully thank you for helping us to improve significantly the scientific content of this manuscript.

Best regards

Dr. A. Boughriet

Villeuve d’Ascq (France), September 29th 2022

To Reviewer 3

Dear Madam or Sir,

We have revised the manuscript according Reviewers’ recommendations and suggestions.

We have made the following corrections and modifications (the added text is typed in red).

(1)A discussion on the adsorptive membrane characteristics of the brick composite (with quartz as support matrix) is now made in the introduction section. Recent “REVIEW” references relative to membrane applications are also cited in the text.

(2)A better understanding of the dielectric behaviour of the composite and its contribution to the affinity of Brønsted sites has been addressed in the manuscript on a thermodynamic base of interfacial water/solid phenomena developed by James and Healy. In addition, to confirm the existence of bridging Brønsted acid sites at zeolite surfaces, 1H MAS NMR spectroscopy had been used successfully and NMR data are now given in the text. However, more detailed NMR investigations are still under way to study two-dimensional 1H—29Si, 1H−27Al and 29Si−27Al dipolar correlations (D-HMQC) NMR.

(3)XRD measurements were previously performed on composite samples. A reference is cited in the text and a brief comment on obtained XRD patterns is made in this new version.

(4)References relative to XRD characteristics of NaA and NaP are now cited in the text. Note that XRD analysis permitted us to evidence the predominance of NaA crystals to NaP crystals.

(5)English grammar has been checked at best.

We gratefully thank you for helping us to improve significantly the scientific content of this manuscript.

Best regards

Dr. A. Boughriet

Villeuve d’Ascq (France), September 29th 2022

To Reviewer 3

Dear Madam or Sir,

We have revised the manuscript according Reviewers’ recommendations and suggestions.

We have made the following corrections and modifications (the added text is typed in red).

(1)A discussion on the adsorptive membrane characteristics of the brick composite (with quartz as support matrix) is now made in the introduction section. Recent “REVIEW” references relative to membrane applications are also cited in the text.

(2)A better understanding of the dielectric behaviour of the composite and its contribution to the affinity of Brønsted sites has been addressed in the manuscript on a thermodynamic base of interfacial water/solid phenomena developed by James and Healy. In addition, to confirm the existence of bridging Brønsted acid sites at zeolite surfaces, 1H MAS NMR spectroscopy had been used successfully and NMR data are now given in the text. However, more detailed NMR investigations are still under way to study two-dimensional 1H—29Si, 1H−27Al and 29Si−27Al dipolar correlations (D-HMQC) NMR.

(3)XRD measurements were previously performed on composite samples. A reference is cited in the text and a brief comment on obtained XRD patterns is made in this new version.

(4)References relative to XRD characteristics of NaA and NaP are now cited in the text. Note that XRD analysis permitted us to evidence the predominance of NaA crystals to NaP crystals.

(5)English grammar has been checked at best.

We gratefully thank you for helping us to improve significantly the scientific content of this manuscript.

Best regards

Dr. A. Boughriet

v

Round 2

Reviewer 1 Report

Reviewers greatly appreciate the efforts that have been made by the author to improve the quality of their articles after peer review. I reread the author's manuscript and further reviewed the changes made along with the responses from previous reviewers' comments. Unfortunately, the authors failed to make some of the substantial improvements they should have made making this article not of decent quality with biased, not cutting-edge updates on the research topic outlined. In addition, the author also failed to address the previous reviewer's comments, especially on comments number 5, 6, and 8. With all due respect, the reviewer opposed this article to be published and must be rejected. Thank you very much for the opportunity to read the author's current work. 

Author Response

Villeuve d’Ascq ‘France), October 14th 2022

To Reviewer 1

Dear Madam or Sir,

We have revised the manuscript according your recommendations and suggestions.

We have made the following corrections and modifications (the added text is typed in red).

  1. In the introduction section, more details and explanations about novelty in the present study are given in this new version. More (recent) references are also cited in the text for a better understanding of our scientific objectives and future perspectives.

  1. There is still a clear research gap between structure and surface characteristics of adsorptive composite membrane (surface area, porous structure, permeability, selectivity) and membrane capacitance / conductance (related to protons affinity / reactivity. While the FEM analysis should effectively serve as the connection between them and help to conceive artificially new membranes for resolving specific adsorption problems.

For that purpose, determination of thermodynamic constants for surface protonation reactions taking place in brick composite then appear preliminarily very important for a better understanding of surface phenomena inside membrane toward specific cations.

 7 and 8  Detailed comments and recent references concerning computational simulation using finite element analysis to study materials performance are now given in this new version. Note that the suggested reference (Ammarullah et al., 2022) is also cited in the text.

Some comments about future perspective are made in the conclusion section.

We gratefully thank you for helping us to improve significantly the scientific content of this manuscript.

Best regards

Dr. A. Boughriet

Reviewer 2 Report

After the appropriate modification of the manuscript,  I recommend the acceptance of the work in its present form.

Author Response

No  supplementary suggestions and criticisms made by Reviewer 2.

Reviewer 3 Report

The authors have revised the manuscript accordingly.

It is recommended to be accepted.

Author Response

No  supplementary suggestions and criticisms made by Reviewer 3.